# Relationship between Cognitive Reserve and Cognitive Impairment in Autonomous and Institutionalized Older Adults

**DOI:** 10.3390/ijerph17165777

**Published:** 2020-08-10

**Authors:** Marina Wöbbeking-Sánchez, Beatriz Bonete-López, Antonio S. Cabaco, José David Urchaga-Litago, Rosa Marina Afonso

**Affiliations:** 1Facultad de Psicología, Universidad Pontificia de Salamanca, 37002 Salamanca, Spain; mwobbekingsa@upsa.es (M.W.-S.); asanchezca@upsa.es (A.S.C.); jdurchagali@upsa.es (J.D.U.-L.); 2Departmento de Psicología de la Salud, Universidad Miguel Hernández de Elche, 03202 Elche, Spain; 3Department of Psychology and Education, University of Beira Interior, 6201-001 Covilhã, Portugal; rmafonso@ubi.pt; 4CINTESIS, Faculty of Medicine, University of Porto, Rua Doutor Plácido da Costa, 4200-450 Porto, Portugal

**Keywords:** cognitive reserve, active aging, cognitive impairment, education level

## Abstract

It is necessary to determine which variables help prevent the presence of decline or deterioration during the aging process as a function of advancing age. This research analyses the relations between cognitive reserve (CR) and cognitive impairment in 300 individuals. It also aims to confirm the influence of different variables (gender, age, level of studies and institutionalization) in CR and in deterioration in a population of older adults. The results indicate that people with higher CR present less deterioration. Regarding the role of the sociodemographic variables in the level of deterioration and CR, there are no differences between men and women, but there are differences in the variables age, level of studies and institutionalization, in such a way that the older age the greater the cognitive deterioration, the higher the level of studies, the more RC and less deterioration and it was found that the non-institutionalized people present less deterioration and greater CR. It is affirmed that two people with similar clinical characteristics may present different levels of pathology, being the CR the explanation of this fact. The results obtained allow us to affirm that the measurement of CR is considered an essential variable for the diagnosis of neurodegenerative diseases.

## 1. Introduction

Aging is a universal process because it affects everyone [1]. According to Sales [2], the aging process is frequently associated with illnesses and deterioration, so it is essential to improve older people’s quality of life. As people age, they suffer a series of losses and decline, especially cognitive deterioration.

Given the mnesic involvement in the aging process and in view of the need intervene; the concept of cognitive reserve (CR) emerged. CR is defined by Stern [3,4] as the capacity to delay age-related cognitive impairment; that is, a higher probability of tolerating the effects of deterioration-related pathology. In other words, it is the capacity to support a greater amount of neuropathology before reaching the threshold where clinical symptomatology begins to manifest [5]. The concept of CR arose from the so-called “Nun study” by Snowdon, which confirmed that there was no direct relationship between the degree of neuropathology and the clinical expression of dementia during a patient’s lifetime [6]. It should be considered that the hypothetical construct of CR is related to flexibility, efficiency and effectiveness in the use of basic and higher cognitive processes [7].

To understand this concept, Stern [3,4] developed two theoretical models to explain the study of CR. First, a passive model, which refers to the brain reserve, related to the person’s biologic skills, and which, among others, corresponds to brain size, number of neurons or synaptic density—variables that facilitate compensation for possible damage caused by degenerative diseases of the central nervous system [8]. Second, the active model, which encompasses two kinds of reserve, Compensation and CR. Compensation alludes to overcoming or reducing losses or deficits through different mechanisms and processes [9]. The central concept of this article, CR, is used to indicate that there is no direct relationship between the degree and amount of brain pathology and the manifestation and intensity of clinical symptoms.

Due to the novelty of the term, a bibliometric analysis was made of the term CR, yielding a small number of publications that consider it a holistic term for a protective factor against cognitive impairment, which encompasses many cognitive, social and emotional aspects. In this work, the relevance of examining CR in the past years and its role in the research of aging is evident [10]. A scientific review, analyzing 564 publications, suggests that individual differences in ability to cope with pathologies are consistent with predicting that people with higher CR. In addition, it is not a fixed construct and continues to evolve by measuring different variables to maintain cognitive function in old age [11].

As has been confirmed, there are various indicators that influence the CR, but some variables, such as gender, age, institutionalization, among others, facilitate its evaluation and may influence the aging process [12]. In addition, according to various authors, CR is not stable over the entire lifetime and depends on several factors, for example, genetic and physical condition, socioeconomic level, education, professional occupation or leisure activities, among others [13]. Many variables or indicators affect the development, maintenance and/or enhancement of CR over one’s lifetime, as it is dynamic and multifactorial [14]. Below, we describe the variables/indicators that have been studied for their influence on CR:

Educational level: the variable that determines CR is education, therefore it is the most studied [15]. One of the existing hypotheses indicates that education may protect against neurocognitive changes, favoring synaptic growth [16].

Intelligence: It is imperative to refer to the role of intelligence in the study of CR.

Different authors have stated that intellectual capacity is one of the variables that largely determine CR and it is therefore essential to evaluate it [17].

Occupation: This refers to the profession carried out during the person’s lifetime. Some authors claim that senior or superior posts are associated with a 50% reduction in the risk of developing dementia [18].

Leisure activities: Another variable that presents evidence of its effects on CR is leisure activities [19].

Other variables: Bilingualism and nutrition [20] are included as influential variables of CR.

Therefore, we can affirm that CR is a multidimensional and multidisciplinary concept. Although research has mainly been carried out in the gerontological field for the prevention of dementia [21], it has generated a growing interest in psychiatry and psychopathology, applying CR in strokes [22], mood disorders [23] or schizophrenia [24].

The objective of this work is to analyze the influence of CR in the cognitive status of older adults, both autonomous and institutionalized. Based on the above literature, we expect to find that the greater the CR, the lower will be the cognitive impairment presented. In addition, we wish to confirm the relationship of the variables gender, age, level of studies and institutionalization with on the level of cognitive impairment and of CR.

## 2. Materials and Methods

This is a non-experimental, cross-correlational study. In it, CR variables and cognitive impairment are evaluated in two samples of older people, a group of institutionalized people (who live in a residential center) and another group who are not institutionalized and who study at university.

### 2.1. Participants

The study included a total of 300 participants, 224 females (75%) and 76 males (25%). Of the total sample, 150 participants were living in institutionalized centers and 150 were not institutionalized. The selection of the simple was selected random. The total sample was made up of adults aged between 55 and 99 years, with a mean age of 74.66 (*SD* = 10.01) for males and of 74.70 (*SD* = 11.62) for females. The mean age for the total simple was 74.76 years.

Regarding the level of studies of the total sample, 10% had no schooling, bordering on illiteracy (n = 30), 51% had primary education (n = 148), 20% had secondary studies (n = 60) and 19% had university studies (n = 58).

The inclusion criteria for both groups were the same, first being 55-year-old or older, not showing cognitive impairment and being institutionalized in a residential center or living at home.

### 2.2. Instruments

*Cuestionario de reserva cognitiva (CCR:* cognitive reserve questionnaire) [25]. This questionnaire evaluates CR through a series of variables (parents’ schooling, the respondent’s educational level, training courses and work occupation, musical training and the knowledge of languages). The CCR is the only questionnaire with normative values for cognitively healthy Spanish population. In this questionnaire, the higher the score, the higher the CR.

The *escala de reserva cognitiva (ECR*: cognitive reserve scale) [26,27]. It measures diverse cognitive domains, such as visuospatial/executive capacity, attention, denomination, language, abstraction, learning, deferred memory and orientation. It presents good reliability (Cronbach’s alpha:.80) and good validity.

The *Montreal cognitive assessment (MoCA)* [28,29] which was designed as a quick screening tool to assess mild cognitive impairment. It measures cognitive domains such as visuospatial/executive capacity, attention, naming, language, abstraction, learning, delayed memory and orientation. In this questionnaire, for higher score, lower cognitive deterioration.

The memory alteration test *(M@T)* [30] is a valid screening test to detect mild cognitive impairment of amnesic type and mild Alzheimer’s disease (AD) [31]. In a similar study [32], sensitivity (≥98%) and specificity (≥97%) were obtained, findings that confirm the validity and efficacy of the present test.

### 2.3. Procedure

This research was carried out in two phases. In the first phase, we contacted the residential centers in order to recruit the sample.

In the second phase, we provided the information about the study to all the participants prior to requesting their informed consent, ensuring the confidentiality of the data. Participants were then assessed. A semi-structured interview was conducted in which data about their sociodemographic data were gathered and next, the battery of tests was administered. The assessment was administered individually to each participant, lasting an hour and a half (March to October of 2017).

Before administering the questionnaires, participants signed an informed consent informing them that their data would be treated confidentially. In this study, invasive procedures are not used, so approval of the bioethics committee was not required. All procedures followed were in accordance with the ethical standards of the committee responsible for human experimentation (institutional and national) and with the Helsinki Declaration of 1975, as revised in 2013.

### 2.4. Data Analysis

We used the Pearson linear correlation to examine the correlations between the quantitative variables. For the study of the differences of stockings for two groups the test T is calculated, ANCOVA and the ANOVA test when there are three groups. The size of the effect was calculated with the Cohen test D. For the case in which the assumption of variance homogeneity is not fulfilled, the robust Welch test and the intergroup post hoc differences with the Tamhane test, is calculated.

## 3. Results

First, we present the descriptive results for each target variable of the study and describe how these variables are distributed in the sample. Then we present the study of relationships between them (Table 1).

In the CCR, 33% of the participants were located in the area of low CR (score of 6 points or less on the scale), 19.3% in the area of medium-low range, 31.7% in the medium-high range and 16% in the area of high CR. This sample presents a mean of 9.49 (*SD* = 4.61).

In the MoCA, 65.3% of the sample presented deterioration and in 34.7%, there was no deterioration (26 points or higher on the scale). The sample mean in this test was 23.1 (*SD =* 4.32). Regarding the M@T, 36.3% was in area of memory alteration (score of 36 or lower) and 63.7% did not present alteration. The sample mean in this test was 37.64 (*SD* = 8.39).

As shown in Table 2, the results indicate that the two CR variables correlated significantly and positively with both measures of memory. Specifically, the CCR correlated with the MoCA (*r* = 0.671) and with the M@T (*r* = 0.696). The ECR also correlated with the MoCA (*r* = 0.576) and with the M@T (*r* = 0.628). (All *p* < 0.001.) Thus, a higher score obtained in the two measurements of CR was related to less memory alteration. As can be seen, the hypothesis of the study was confirmed.

Second, we analyzed the relation with of the variables gender, age, level of studies and institutionalization.

To analyze the influence of the variable gender, we used Student’s t-test of differences of means. There were only significant differences in the MoCA (*p* = 0.034 < 0.05, effect size = 0.27 Cohen’s *d*); no significant differences were found in the rest of the tests (*p* < 0.05). This indicates that there are no differences between men and women in the level of deterioration or in CR. This shows that the hypothesis (No differences are expected between men and women) was confirmed except for the measurement of cognitive impairment as assessed by the MoCA (Table 3).

We examined the correlations of the variable age with the measures of impairment and CR. All of the correlations obtained were significant (*p* < 0.01), in the predicted direction. At greater age, greater cognitive impairment and less RC (Table 4).

Regarding the variable level of studies, as a function of the wording of this hypothesis, the results are presented separately—on one hand, deterioration and on the other hand, CR.

*Cognitive impairment*. The test of homogeneity of variance was significant for all variables (*p* < 0.01), so we cannot assume that the variables are homogeneous in the different groups. Thus, in addition to the ANOVA, we studied the differences with the robust Welch’s test, finding all of them to be significant (*p* < 0.001). This implies that at least one group differs significantly from the rest of the groups in cognitive impairment (Table 5).

The results show that the higher the level of studies, the lower the cognitive impairment and these differences were revealed at all educational levels (*p* < 0.05) (Table 6).

*Cognitive reserve.* Next, we analyzed the relationship between educational level and CR. Significant differences were observed in both indicators of CR (CCR and ECR) and in the predicted direction (ANOVA and WELCH: *p* < 0.001), indicating that a higher level of studies is related to higher CR. In the case of CCR, significant differences were obtained between the four educational levels, whereas in the ECR, these differences between the groups of secondary education and University studies were nonsignificant (but they were significant in the rest of the educational groups).

Lastly, we analyzed the difference in the CR and level of cognitive impairment in institutionalized groups versus non-institutionalized groups. For this purpose, Student’s t-test was performed, yielding significant differences in all of the variables, with high effect sizes (effect size > 0.80).

Finally, we analyzed the difference in CR and the level of cognitive impairment in institutionalized versus non-institutional groups, controlling the effect of studies and age. For this purpose, ANCOVA was performed. Differences between estimated means (after controlling for the effect of the study variable) yielding significant differences in all of the variables (*p* < 0.001), with large effect sizes for RC (ECR y CCR) (ƞ_p_^2^ > 0.15) and medium effect sizes (ƞ_p_^2^ > 0.06) for Impairment (MoCA y M@T) (Table 7).

In terms of tests indicating *deterioration* (MoCA and M@T), significant differences were found between the institutionalized and non-institutionalized groups, in such a way that the institutionalized group scores higher in both impairment tests. Significant differences were also found in CR, so that the non-institutionalized group scored higher in both CRC and ECR than the institutionalized group.

## 4. Discussion

The goal of this research was to examine the influence of CR in the participants’ cognitive status and to analyze the independent variables. The results are in the line of others studies that [33] supporting the hypothesis that CR influences the delay of the manifestation of symptoms of cognitive impairment. Others authors [18,34,35] also pointed out the existence of numerous works that claim that the manifestations of pathologic symptoms are delayed in people who have a high CR. Recently, in a theoretical review [14] claimed that people with high CR present a slower progression of cognitive decline than people with low CR.

Regarding the cognitive level, the scientific literature has shown the existence of a relationship between the level of cognitive performance and CR, both in healthy individuals and in people with different cognitive pathologies [36]. Likewise, others authors [37] Mayordomo, Sales and Meléndez (2015) state that CR can be considered a protector factor against the expression of age-related cognitive impairment.

We use the variable “age” of the present work to confirm that older age is related to lower CR. In an important investigation [38] In a García-Sevilla, Fernández, Fuentes, López and Moreno (2014) stated that, over time, there is a decline and a slowing down of cognitive abilities. In this same direction, others authors [39,40] Pankratz et al. (2015) and Huxhold, Miche and Schüz (2014) indicated that the variable age is crucial to the differences among older adults, not only to confirm whether cognitive performance is better or worse, but also to compare young people, adults or older adults.

Third, many researchers believe that educational level is a very powerful indicator of CR, as it was found to have direct influence on aging and on the development of cognitive impairment [41] (Volpi et al. 2017). Vásquez-Amézquita, in 2016 [42] stated that education is one of the strongest predictors of CR, whereas other studies reaffirm that education not only affects CR, but also the brain reserve, as an association was found between the level of education, brain volume and educational performance.

Finally, in relation to the sixth variable, institutionalization, the hypothesis was confirmed in the present study, therefore, the non-institutionalized group presented less deterioration and greater CR than in the institutionalized group. Regarding the association between institutionalized individuals and the prevalence of cognitive impairment in older adults, in a cross-sectional descriptive observational study, Vallejo and Rodríguez [43] found that 27% of the institutionalized old adults presented cognitive impairment. The results obtained in this study highlight the existence of differences between the groups of institutionalized and non-institutionalized adults.

In view of these findings, we can state that two people with similar clinical features may have different levels of pathology and that CR may be responsible for this fact. Therefore, in view of the results obtained in this study, it can be claimed that the measure of CR should be taken into account for the diagnosis of neurodegenerative diseases, as well as to determine possible treatments [44].

We want to make explicit some aspects to improve this work. In terms of the size and characteristics of the sample, the role of variables such as the context (in this work, the sample is urban) should be explored, as well as the role of the residency setting, due to its association with other variables linked to key cultural and stimulation opportunities. Closely related to this issue is the type of residence (public–private), our sample only included older individuals in private or concerted residences of a medium-high economic level. Regarding active and passive models of CR, it would be of interest to confirm the combination of biologic indicators [1] with the measures used in this study. The possible correlation between the two would show the correspondence between structural and functional components, which should be considered to optimize healthy human longevity.

## 5. Conclusions

From the objectives formulated and the analysis of the results, it is concluded that those people, both institutionalized and community dwelling, with high scores in cognitive reserve have less cognitive deterioration. It would be convenient to carry out an experimental design that would allow us to draw causal conclusions and control the effect of strange variables

The present work shows the important relationship established between cognitive impairment and cognitive reserve, the latter being a key factor in the process of preventing pathologic aging.

## Figures and Tables

**Table 1 ijerph-17-05777-t001:** Means, standard deviations and standard mean error in the instruments used.

	Instrument			
	ECR	CCR	MoCA	M@T
Mean	43.07	9.49	23.13	37.64
SD	13.75	4.61	4.32	8.39

Note: ECR—escala de reserva cognitiva [cognitive reserve scale]; CCR—cuestionario de reserva cognitiva [cognitive reserve questionnaire]; MoCA—Montreal cognitive assessment; M@T—memory alteration test.

**Table 2 ijerph-17-05777-t002:** Pearson correlations between cognitive reserve (CCR and ERC) and cognitive level (MoCA and M@T).

Cognitive Functioning	MOCA	M@T
ERC	0.576 ***	0.628 ***
CCR	0.671 ***	0.696 ***

Note: ECR—escala de reserva cognitiva [CR Scale]; CCR—cuestionario de reserva cognitiva [CR questionnaire]; MoCA—Montreal cognitive assessment; M@T—memory alteration test; *** *p* < 0.001 (two-tailed).

**Table 3 ijerph-17-05777-t003:** Sample descriptives using t-test.

		Sex		*t*-Test (*p*)	Effect Size Cohen’s d
	Instrument	Male *M* (*SE*)	Female *M* (*SE*)		
CR	ECR	41.0 (15.9)	43.8 (12.9)	0.18	−0.19
	CCR	9.9 (4.59)	9.4 (4.61)	0.42	0.12
Impairment	MoCA	23.9 (3.82)	22.8 (4.44)	0.03	0.27
	M@T	38.5 (7.79)	37.3 (8.57)	0.28	0.15

Note: ECR—escala de reserva cognitiva [CR Scale]; CCR—cuestionario de reserva cognitiva [CR questionnaire]; MoCA—Montreal cognitive assessment; M@T—memory alteration test.

**Table 4 ijerph-17-05777-t004:** Correlations between age and cognitive reserve.

	Age	
Instrument	Pearson Correlation	Sig. (2-Tailed)
ECR	−0.49 ^***^	<0.0001
CCR	−0.48 ^***^	<0.0001
MoCA	−0.54 ^***^	<0.0001
M@T	−0.62 ^***^	<0.0001

Note: ECR = escala de reserva cognitiva [CR Scale], CCR = cuestionario de reserva cognitiva [CR questionnaire], MoCA = Montreal cognitive assessment, M@T = memory alteration test *** *p* < 0.001 (2-tailed).

**Table 5 ijerph-17-05777-t005:** ANOVA and descriptive statistics on deterioration.

		Level of Studies				
Instrument		NS	PE	SE	UN	Comparison between Groups ^1^
MoCA	*M*	19.7	21.7	25.1	26.5	NS<PE*; NS< SE and UN ***
	*SD*	3.47	4.36	2.48	2.66	PE< SE and UN ***
						SE < UN *
M@T	*M*	29.1	34.9	42.2	44.5	NS<PE **; NS< SE and UN ***
	*SD*	7.95	7.97	5.06	3.59	PE< SE and UN ***
						SE < UN *

Note: NS—no studies, PE—primary education, SE—secondary education, UN—university studies. MoCA—Montreal cognitive assessment, M@T—memory alteration test.^1^ post hoc tests—Tamhane for the assumption of nonhomogeneous variances.* *p* < 0.05. ** *p* < 0.01. *** *p* < 0.001.

**Table 6 ijerph-17-05777-t006:** ANOVA and descriptive statistics on deterioration cognitive reserve.

Instruments		Level of Studies				Comparison between Groups ^1^
		NS	PE	SE	UN	
ECR	*M*	28.4	38.9	50.7	53.6	NS<PE, SE and UN ***
	*SD*	5.43	11.96	9.67	12.95	PE< SE and UN ***
						SE = UN
CCR	*M*	3.4	7.4	12.1	15.4	NS<PE, SE and UN ***
	*SD*	1.43	2.98	2.60	2.47	PE< SE and UN ***
						SE < UN **

Note: NS—no studies; PE—primary education; SE—secondary education; UN—university studies; Note—ECR—escala de reserva cognitiva [CR Scale]; CCR—cuestionario de reserva cognitiva [CR questionnaire]; ^1^ post hoc tests—Tamhane for the assumption of nonhomogeneous variances. ** *p* < 0.01. *** *p* < 0.001.

**Table 7 ijerph-17-05777-t007:** Relationship of CR and level of cognitive impairment in institutionalized vs. non-institutionalized people. Means and Standard Deviations corrected according to studies and age (ANCOVA).

Cognitive Functioning		Institutionalized		*F*(3, 296)	*p*	ƞ_p_^2^
		Yes *M* (*SE*)	No *M* (*SE*)			
CR	ECR	34.3 ^a^ (1.0)	51.9 ^a^ (1.0)	126.8	<0.001	0.263
	CCR	7.9 ^a^ (0.26)	11.1 ^a^ (0.26)	290.0	<0.001	0.255
Impairment	MoCA	22.1 ^a^ (0.36)	24.1 ^a^ (0.36)	78.0	<0.001	0.036
	M@T	34.4 ^a^ (0.80)	40.9 ^a^ (0.59)	143.1	<0.001	0.122

Note: ECR—escala de reserva cognitiva [CR Scale]; CCR—cuestionario de reserva cognitiva [CR questionnaire]; MoCA—Montreal cognitive assessment; M@T—memory alteration test. ^a^ Covariates appearing in the model are evaluated at the following values: Age = 74.69, Level of studies = 2.49.

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
