# Peer review of "Relationship between Cognitive Reserve and Cognitive Impairment in Autonomous and Institutionalized Older Adults"

_ijerph, 2020, doi:10.3390/ijerph17165777_

Round 1

Reviewer 1 Report

I think that this manuscript was poorly prepared. I would suggest for the authors to carefully revise the manuscript and consider re-submission. Consideration should be given to spelling errors, grammar, syntax, and expressions used in English. The authors may also consider to describe thoroughly in the text what cognitive reserve is, and what their study adds to the existing knowledge on the topic of cognitive ageing and cognitive reserve. 

Author Response

Por favor vea el archivo adjunto 

Reviewer 2 Report

The authors are interested in the relations between Cognitive Reserve (CR) and cognitive impairment in 300 individuals and whether the influence of different variables (gender, age, level of studies, and institutionalization) confirm deterioration in a population of older adults.

There is one crucial issue that has to be clarified:

On page 2, under Materials and Methods the authors state that “CR variables and cognitive impairment are evaluated in two samples of older people, a group of institutionalized people (who live in a residential center) and another group who are not institutionalized and who study at university.” Was that really the case? As they could confirm in their data, education level was significantly influencing CR. Therefore, the above mentioned dichotomization would be substantially biased by education level. Thus, I strongly suggest to compare CR of institutionalized and non-institutionalized people controlled for education level and age. Otherwise, results are – as described above - substantially confounded.

I have only minor suggestions:

Please change Spanish words with English in tables 5 and 6 “Comparison between groups” (“and” instead of “y”)

Author Response

Por favor vea el archivo adjunto

Round 2

Reviewer 2 Report

No more comments